# Enhancing Antimicrobial Peptides from Frog Skin: A Rational Approach

**DOI:** 10.3390/biom15030449

**Published:** 2025-03-20

**Authors:** Silvana Aguilar, Daniel Moreira, Ana Laura Pereira Lourenço, Natalia Wilke, Matías A. Crosio, Andreanne Vasconcelos, Eder Alves Barbosa, Elizabete C. I. Bispo, Felipe Saldanha-Araujo, Marcelo H. S. Ramada, Franco M. Escobar, Cristina V. Torres, José R. S. A. Leite, Mariela M. Marani

**Affiliations:** 1IPEEC-CONICET, Consejo Nacional de Investigaciones Científicas y Técnicas, Puerto Madryn U9120ACD, Argentina; saguilar@cenpat-conicet.gob.ar; 2Research Center in Morphology and Applied Immunology, NuPMIA, Faculty of Medicine, University of Brasília, UnB, Brasília 70910-900, DF, Brazil; moreiradc@unb.br (D.M.); andreannegv@gmail.com (A.V.); bioederr@gmail.com (E.A.B.); jrsaleite@gmail.com (J.R.S.A.L.); 3Programa de Pós-Graduação em Ciências Genômicas e Biotecnologia, Universidade Católica de Brasília, Taguatinga 71966-700, DF, Brazil; analaurapl95@gmail.com (A.L.P.L.); marcelo.ramada@p.ucb.br (M.H.S.R.); 4Facultad de Ciencias Químicas, Departamento de Química Biológica Ranwel Caputto, Universidad Nacional de Córdoba, Córdoba X5000HUA, Argentina; natalia.wilke@unc.edu.ar (N.W.); matias.crosio@unc.edu.ar (M.A.C.); 5Centro de Investigaciones en Química Biológica de Córdoba (CIQUIBIC), CONICET, Universidad Nacional de Córdoba, Córdoba X5000HUA, Argentina; 6University Center of the Federal District, UDF, Brasília 70390-045, DF, Brazil; 7Laboratory of Synthesis and Analysis of Biomolecules, LSAB, Institute of Chemistry, IQ, University of Brasília, UnB, Brasília 70910-900, DF, Brazil; 8Laboratory of Hematology and Stem Cells, Faculty of Health Sciences, University of Brasília, UnB, Brasília 70910-900, DF, Brazil; elizabete.iseke@hotmail.com (E.C.I.B.); felipearaujo@unb.br (F.S.-A.); 9Departamento de Microbiología e Inmunología, Universidad Nacional de Río Cuarto, Ruta 36, Km 601, Río Cuarto 5800, Argentina; fescobar@exa.unrc.edu.ar (F.M.E.); ctorres@exa.unrc.edu.ar (C.V.T.)

**Keywords:** hylins, in silico design, amphipathic, peptide–membrane interactions, cell selectivity

## Abstract

Antimicrobial resistance is a global health threat, which has been worsened by the slow development of new antibiotics. The rational design of natural-derived antimicrobial peptides (AMPs) offers a promising alternative for enhancing the efficacy of AMPs and accelerating drug discovery. This paper describes the rational design of improved peptide derivatives starting from hylin-Pul3, a peptide previously isolated from the frog *Boana pulchella,* by optimizing its hydrophobicity, cationicity, and amphipathicity. In silico screening identified six promising candidates: dHP3-31, dHP3-50, dHP3-50.137, dHP3-50.190, dHP3-84, and dHP3-84.39. These derivatives exhibited enhanced activity against Gram-negative bacteria, emphasizing the role of cationicity and the strategic arginine incorporation. Hemolytic assays revealed the derivatives’ improved selectivity, particularly for the derivatives with “imperfect amphipathicity”. In fibroblast assays, dHP3-84 was well-tolerated, while dHP3-84.39 promoted cell proliferation. Antioxidant assays (ABTS assays) highlighted the Trp-containing derivatives’ (dHP3-50.137, dHP3-31) significant activity. The lipid membrane interaction studies showed that hylin-Pul3 disrupts membranes directly, while dHP3-84.39, dHP3-50, and dHP3-50.137 promote vesicle aggregation. Conversely, dHP3-84 did not induce membrane disruption or aggregation, suggesting an intracellular mode of action. Machine learning models were effective in predicting bioactivity, as these predicted AMPs showed enhanced selectivity and potency. Among them, dHP3-84 demonstrated broad-spectrum potential. These findings highlight the value of rational design, in silico screening, and structure–activity studies in optimizing AMPs for therapeutic applications.

## 1. Introduction

Antimicrobial resistance (AMR) is a critical global health threat that endangers both public health and modern medicine, accounting for millions of deaths annually [1]. The emergence, widespread prevalence, and rapid dissemination of AMR have been further aggravated by the sluggish development of new antibiotics [2]. In response, the World Health Organization (WHO) has emphasized the urgent need to address resistant Gram-negative pathogens. To tackle this growing challenge effectively, the WHO advocates for increased research investment and a holistic approach that integrates human, animal, and environmental health [1].

Natural peptides are re-emerging as promising candidates for development as innovative therapeutic agents [3], with many currently undergoing clinical evaluation for their potential applications [4]. Antimicrobial peptides (AMPs) are widely distributed across the natural world, with over a thousand identified specifically in amphibians. These peptides primarily exert their effects by targeting and disrupting membrane stability and function. Additionally, they interfere with cytoplasmic components, inhibiting essential processes such as cell wall synthesis, protein synthesis, and enzymatic activity. These multiples modes of action make AMPs less likely to induce resistance [2,5].

Amphibian-derived AMPs share key structural features, including cationicity, amphipathicity, and α-helical conformations, which are crucial for their antibacterial activity [6,7]. In addition to their potent antibacterial properties, these peptides offer promising opportunities for optimizing efficacy across a broad range of applications, including antioxidant, antiviral, and anticancer therapies [8,9]. The modularity and adaptability of AMPs isolated from frog skin make them excellent scaffolds for drug design and optimization.

Rational design has become a cornerstone in the development of AMPs, providing a systematic approach to modifying natural peptides to improve their efficacy and safety. Structure–activity relationship (SAR) studies play a central role in guiding these modifications, allowing researchers to analyze how specific structural features—such as hydrophobicity, helicity, cationicity, and amphipathicity—affect the bioactivity of AMPs [10,11]. By leveraging SAR insights, AMPs can be tailored to enhance selectivity, reduce cytotoxicity, improve stability, and overcome key challenges in therapeutic peptide development [12]. Recent advances have shown that modifying the primary sequence of AMPs to increase cationicity, helical propensity, hydrophobicity, and amphipathicity can achieve a delicate balance between antimicrobial potency and biocompatibility, a critical factor for their safe and effective therapeutic use [13,14,15,16].

The use of different bioinformatics tools requires a database of peptides with known antimicrobial activities [17]. Expanding the number of peptides in such models enhances the models’ robustness and predictive utility [18]. Computational methods enable the extraction of key structural and biophysical features from extensive databases, facilitating the prediction and enhancement of peptide antimicrobial activity [19]. Nevertheless, experimental validation remains indispensable for confirming activity, assessing cytotoxicity, and elucidating the mechanisms of action.

In this paper, we report the rational design of improved peptide derivatives using hylin-Pul3, a peptide isolated from the frog *Boana pulchella*, as the template sequence. Hylin-Pul3 (FLGALIPAIAGAIGGLIRK-NH_2_) has shown potent activity against *S. aureus* (MIC 14 μM) and moderate activity against *E. coli* (MIC 108 μM), but it also exhibited significant hemolytic activity at concentrations effective against Gram-negative bacteria [20]. To improve its therapeutic profile, we employed an integrated approach, combining in silico design and in vitro validation, to develop hylin-Pul3 derivatives with enhanced bioactivity and reduced cytotoxicity. Specifically, we assessed the antibacterial, antioxidant, antiviral, and cytotoxic properties of these derivatives and investigated their interactions with lipid membranes to elucidate their mechanisms of action.

## 2. Materials and Methods

### 2.1. Rational Design of Derivative Peptides and Bioinformatic Analysis

Starting from the parent peptide hylin-Pul3, a series of derivative peptides were designed through rational modifications of the primary sequence. The design process was guided by the following criteria: (i) a net charge greater than +2, (ii) a predicted α-helix content exceeding 50%, and (iii) adherence to specific sequence restrictions, such as the exclusion of certain residues due to the challenges they can pose during synthesis (e.g., Cys or Met at any position due to their susceptibility to oxidation, and Asn at the N-terminal to avoid cyanoalanine formation). Additionally, specific amino acid pairings, such as Asp-Gly, Asp-Pro, and Asp-Ser, were intentionally avoided to prevent aspartimide formation [21]. Strategic modifications to the primary sequence were implemented to achieve desired properties, including reducing the sequence length, altering the hydrophobicity, enhancing the cationic properties, evaluating the charge distribution, creating structures with both perfect and imperfect amphipathicity and incorporating aromatic amino acids.

To characterize and evaluate the designed peptide derivatives, bioinformatic analyses were conducted using the following established online software tools (Figure 1): (i) ProtParam [22] for the molecular weight, pI, charge, and grand average of hydropathicity (GRAVY) analyses; (ii) HNN (Hierarchical Neural Network) [23] for secondary structure prediction; (iii) HeliQuest v1.2 online [24] for the hydrophobicity (<H>), hydrophobic moment (µH), and helical wheel projection analyses; (iv) AlphaFold2 for 3D structure prediction, visualized with UCSF Chimera v1.13.1 [25]; (v) CAMP_R3_ [26,27] for the antimicrobial activity probability analysis; (vi) the Database of Antimicrobial Activity and Structure of Peptides (DBAASP) platform v3 [28] for antibacterial activity prediction; and (vii) HAPPENN [29] and HemoPred [30] for hemolytic effect estimation. The steps employed to select derivative peptides for synthesis and the bioactivity assays are summarized in Figure 1.

### 2.2. Synthesis and Purification of Derivative Peptides

#### 2.2.1. Solid-Phase Peptide Synthesis (SPPS)

The selected peptides were manually synthesized using solid-phase peptide synthesis (SPPS) with Fmoc/tert-butyl chemistry [31]. Microwave-assisted synthesis was performed on Wang resin (Peptides International; Louisville, KY, USA; substitution 0.6 mmol/g) for C-terminal carboxy peptides and on Rink amide MBHA resin (Peptides International; Louisville, KY, USA; substitution 0.51 mmol/g) for amidated peptides. The resins were preconditioned with dichloromethane (DCM) and dimethylformamide (DMF). For the first amino acid coupling, 3 equivalents (eq) of Fmoc-AA-OH, 4 eq of N,N′-diisopropylcarbodiimide (DIPCDI) as the activator, and 0.1 eq of N,N-dimethyl-4-aminopyridine (DMAP) as a hyper-nucleophilic catalyst were dissolved in minimal DMF. This mixture was subjected to microwave irradiation for 30 s at minimal power (70 W) at room temperature, which was repeated four times. Acetylation was then performed using 6 eq of acetic anhydride (Ac_2_O) with 0.1 eq of DMAP. Peptide elongation was carried out using 3 eq of Fmoc-AA-OH, 3 eq of TBTU, and 6 eq of N,N-diisopropylethylamine (DIPEA) in minimal DMF, with each amino acid incorporated via microwave irradiation. Coupling was confirmed using Kaiser’s test [32]. If the test result was positive, the coupling was repeated. Fmoc deprotection at the N-terminus of each amino acid was achieved using two washes of 20% piperidine in DMF at room temperature, for 10 min each.

Peptides were cleaved from the resin and deprotected by treatment with a trifluoroacetic acid/triisopropylsilane/water (TFA/TIS/H_2_O) solution (95:2.5:2.5, *v*/*v*/*v*). Crude peptides were precipitated with cold ether dissolved in H_2_O:MeCN (1:1) and lyophilized. Additionally, 5 mg of synthetic hylin-Pul3 and dHP3-84 peptides with >95% purity were provided by the company Synpeptide Co., Ltd. (Nanjing, China).

#### 2.2.2. Reverse-Phase High-Performance Liquid Chromatography (RP-HPLC) and Characterization with Mass Spectrometry Analysis

The synthesized peptides were dissolved in H_2_O and purified using a reverse-phase high-performance liquid chromatography (RP-HPLC) system (Shimadzu; Kyoto, Japan). The system comprised a degassing unit (Model DGU-20A5R), a binary pump system with two semipreparative solvent delivery units (Model LC-20AR), a photodiode array (PDA) detector (Model SPD-M20A), and a controller communication bus module (Model CBM-20A). Purification was performed on a C18 column (10 µm pore size, 250 × 21.2 mm), with absorbance monitored in the range of 190–300 nm.

A binary mobile-phase method was used, consisting of solution A (H_2_O with 0.1% (*v*/*v*) TFA) and solution B (MeCN with 0.1% (*v*/*v*) TFA). The flow rate was set at 10 mL·min^−1^. The elution conditions were as follows: 0–5 min at 0% solution B, followed by a linear gradient from 0% to 100% solution B over 5–45 min, and finally 100% solution B from 45 to 55 min. The fractions were manually collected, lyophilized, and subsequently analyzed.

The sequence of the synthetic peptides was confirmed by matrix-assisted laser desorption/ionization time-of-flight tandem mass spectrometry (MALDI-TOF/TOF). The lyophilized fractions were dissolved in a 50% acetonitrile solution, mixed with α-cyano-4-hydroxyxinnamic acid (10 mg/mL dissolved in 50% acetonitrile/0.3% TFA (*v*/*v*) in Milli-Q water) in a ratio of 1:3 (*v*/*v*), spotted onto a MALDI plate, and evaporated at room temperature. Acquisitions and analyses were performed using a MALDI-TOF-AutoFlex instrument (Bruker Daltonics; Bremen, Germany) operating in the positive mode, according to a previously published protocol [33]. Alternatively, dissolved fractions were directly injected (2 µL) into a Maxis 4G mass spectrometer (Bruker Daltonics; Bremen, Germany) using an Acquity chromatographic system (Water; Massachusetts, USA) without a column, an isocratic gradient of 50% acetonitrile containing 0.1% formic acid, and a flow rate of 0.4 mL/min. The mass spectrometer operated in the positive and data-dependent acquisition (DDA) modes. All MS/MS spectra were manually annotated using FlexAnalysis v3.3 or DataAnalysis software (Bruker Daltonics; Bremen, Germany).

### 2.3. Biological Activity

#### 2.3.1. Antibacterial Assays

Hylin-Pul3 and six derivatives (dHP3-31, dHP3-50, dHP3-50.137, dHP3-50.190, dHP3-84, and dHP3-84.39) were tested against two Gram-positive bacteria, *Staphylococcus aureus* (ATCC 25923) and *Streptococcus mutans* (ATCC 25175), as well as three Gram-negative bacteria: *Escherichia coli* (ATCC 25922), *Klebsiella pneumoniae* (ATCC 13883), and *Acinetobacter baumannii* (ATCC 13304). The minimal inhibitory concentration (MIC) was determined using the microbroth dilution [33].

The peptides were serially diluted in concentrations ranging 128 to 0.25 μM, with gentamicin serving as a positive control, which was serially diluted to concentrations of 20 to 0.04 μM. Bacterial cultures were prepared in Mueller–Hinton (MH) medium to achieve a final inoculum concentration of 5 × 10^5^ CFU/mL. The MIC was defined as the lowest peptide concentration that completely inhibited visible bacterial growth. All tests were performed in duplicate and repeated across three independent experiments.

#### 2.3.2. Synergistic Checkerboard Assay

Peptides dHP3-84 and dHP3-50.137, which exhibited the lowest MIC values against *E. coli* (ATCC 25922), were selected to evaluate their synergistic effects [34]. Briefly, working solutions of the peptides were prepared and serially diluted by half across the rows and columns of a microplate, ensuring that each well contained a unique combination of peptide concentrations. The starting concentrations corresponded to the respective MIC values of each peptide, followed by two-fold serial dilutions.

Each well was filled with 50 µL of the peptide solution (individual or combined) and 50 µL of a bacterial suspension (1 × 10^6^ CFU/mL), resulting in a final bacterial concentration of 5 × 10^5^ CFU/mL. The individual peptide solutions served as positive controls, while the MH medium was employed as a negative control. The microplates were incubated at 37 °C overnight. The test was performed in three independent experiments. The fractional inhibitory concentrations index (ΣFIC) was calculated using the following equation:
(1)$$\sum FIC\;=\frac{MIC_{\left(A\right)}\;in\;combination}{MIC_{\left(A\right)}\;alone}+\frac{MIC_{\left(B\right)}\;in\;combination\;}{MIC_{\left(B\right)}\;alone}$$

The FIC index (ΣFIC) was interpreted as follows: ΣFIC ≤ 0.5 indicated synergy; 0.5 < ΣFIC ≤ 4 indicated no effect; and 4 < ΣFIC indicated antagonism.

#### 2.3.3. Antiviral Activity Assays

Antiviral activity assays were performed following a previously described protocol [35] with slight modifications. The efficacy of the peptide compounds was assessed at different stages of the viral lifecycle, including adsorption, penetration, and intracellular replication. The hylin-Pul3 and dHP3-84 peptides were tested at non-cytotoxic concentrations of 20 μg/mL (10.8 μM) and 75 μg/mL (34.5 μM), respectively. Vero cell monolayers (African green monkey kidney cells) were cultured in 24-well plates and exposed to the peptide solutions along with serial 10-fold dilutions of *Varicellovirus suidalpha1*, formerly known as *Herpes suis virus type 1* (HSV-1). The assays were performed in duplicate. After a 90 min incubation at 37 °C, any residual unadsorbed virus was removed, and the monolayers were incubated for an additional 72 h at 37 °C with medium (MEM with 2% FCS) supplemented with the corresponding peptide at the same concentration used during adsorption and penetration. Following the incubation, the cell monolayers were fixed with 10% formalin and subsequently stained to visualize viral plaques. The viral plaques that formed on the Vero cells were stained (1% crystal violet) and the viral titers were determined. The percentage viral inhibition was calculated by comparing the reduction in viral titer between the treatment and control groups. This was determined using the following formula:
(2)% viral inhibition=100−A/B×100 where “A” represents the viral titer expressed in PFU/mL (Plaque-Forming Units per milliliter) obtained in the treated cultures and “B” is the viral titer expressed in PFU/mL obtained in the untreated cultures.

#### 2.3.4. Hemolysis Test

The hemolytic activity of the peptides was assessed using human erythrocytes obtained from a healthy donor, following a previously published protocol [36]. Briefly, equal volumes of erythrocytes were incubated with varying peptide concentrations (1.56 to 200 μg/mL) at 37 °C for 1 h. Hemolysis was determined by measuring the release of hemoglobin after centrifugation (3100× *g* for 5 min at 4 °C). The absorbance of the supernatant was measured at 540 nm to quantify the released hemoglobin. Complete hemolysis (100%) was induced using 0.1% (*v*/*v*) Triton X-100 as the positive control. The percentage of hemolysis was calculated using the following equation:
(3)$$\%\;hemolysis=\frac{A_{\left(peptide\right)}-A_{\left(PBS\right)}\;}{A_{\left(Triton\right)}-A_{\left(PBS\right)}\;}\times 100$$ where “A _(peptide)_” is the absorbance value of the tested solutions, “A _(PBS)_” is the absorbance value of the negative control, and “A _(Triton)_” is the absorbance value of the positive control.

#### 2.3.5. In Vitro Cytotoxicity Against Human Fibroblasts

##### Cell Lines and Cell Culture

Human primary fibroblasts were kindly provided by CellSeq Solutions (Belo Horizonte, Brazil) and cultured in DMEM (Dulbecco’s Modified Eagle’s Medium, Gibco, Waltham, MA, USA) supplemented with 10% (*v*/*v*) fetal bovine serum (FBS; Gibco, Waltham, MA, USA) and a 1000 U/mL penicillin/streptomycin solution (Gibco, Waltham, MA, USA). The cells were maintained at 5% CO_2_, 37 °C, and 95% humidity.

##### Cell Viability and Proliferation

To investigate the impact of the antimicrobial peptides hylin-Pul3, dHP3-31, dHP3-50, dHP3-50.137, dHP3-50.190, dHP3-84, and dHP3-84.39 on primary human fibroblast proliferation and viability, the 3-(4,5-dimethylthiazol-2-yl)-2,5-diphenyl tetrazolium bromide (MTT) metabolization assay (Sigma Aldrich; Misuri, USA) was performed. Human fibroblast cells (7 × 103 cells/well) were plated in a 96-well plate and treated with 0.5, 1, 5, 10, 25, 50, and 100 μg/mL of each peptide for 24 h. After this period, the cells were incubated with 5 mg/mL MTT for 4 h. The formazan crystals formed were solubilized by adding dimethyl sulfoxide (DMSO). The absorbance was measured at 570 nm using a spectrophotometer (Multiskan™ FC Microplate Photometer; Thermo Fisher Scientific, Waltham, MA, USA).

#### 2.3.6. In Vitro Cytotoxicity Against Vero Cells

The cytotoxicity of hylin-Pul3 and dHP3-84 peptides was evaluated on Vero cells procured from the Asociación Banco Argentino de Células (ABAC; Buenos Aires, Argentina). Cell viability was determined using the Neutral Red Uptake test (NRU) test, following a previously described methodology [37]. The Vero cells were cultured in Minimum Essential Medium (MEM) supplemented with 8% fetal calf serum (FCS), 50 μg/mL gentamicin, and 2 mM glutamine. The cells were monitored by observing their morphological changes after 24 and 48 h of incubation.

For the NRU assay, the cells were seeded in 96-well culture plates at a density of 1 × 105 cells per well. The cell monolayers were treated with increasing concentrations of the peptide solutions (25–500 μg/mL) for 48 h in triplicate. The monolayers incubated with MM only were used as the control. Following exposure, the culture medium was replaced with 150 μL of a 50 mg/mL neutral red solution in MEM. After incubation at 37 °C for 3 h, the dye medium was discarded and the wells were gently washed with PBS. The intracellular dye was extracted using a mixture of acetic acid, ethanol, and water (1:50:49). After gentle agitation for 10 min, the absorbance values were measured at 540 nm.

The relative cell viability was calculated as a percentage of the NRU observed in the untreated control groups. The cell survival fraction was determined using the following formula:
(4)$$\mathrm{Survival}\;\mathrm{Fraction}\;\%\;\left(\mathrm{SF}\%\right)=\frac{D.O.\;treated\;cells}{D.O.\;control\;cells}\times 100$$

### 2.4. In Vitro Antioxidant Assays

#### 2.4.1. DPPH Radical-Scavenging Activity

A stock solution of 60 µM 1,1-diphenyl-2-picrylhydrazyl (DPPH) was prepared in ethanol, with its absorbance adjusted to 0.7 at 515 nm immediately before the assay using a Shimadzu 1240 UV–visible spectrophotometer (Kyoto, Japan). The peptides were dissolved in PBS, and 20 µL aliquots of each peptide solution were incubated with 180 µL of the ethanolic DPPH solution in triplicate, at final concentrations of 0.1, 0.05, and 0.025 mg/mL. For the negative control, 20 µL of PBS was mixed with 180 µL of the DPPH solution, while 20 µL of Trolox (6-hydroxy-2,5,7,8-tetramethylchroman-2-carboxylic acid) served as the positive control and standard. The mixtures were incubated in the dark at 25 °C for 30 min, after which, the absorbance was measured at 515 nm. The reduction in absorbance, indicative of DPPH-scavenging activity, was compared to a standard curve. The results were expressed as milligrams of Trolox equivalents per milligram of peptide (mg Trolox equivalents/mg of peptide).

#### 2.4.2. ABTS Assay

The 2,2-azinobis-3-ethylbenzothiazoline-6-sulfonic acid (ABTS^+^) free radical-scavenging activity was determined following the method in [38]. The ABTS⁺ radical cation was generated by diluting a 245 mM ammonium persulfate (APS) solution at a ratio of 1:100 with a 7 mM ABTS solution, followed by overnight incubation in the dark at room temperature. The ABTS^+^ solution was then diluted with water to obtain an absorbance of 0.70 at 734 nm immediately before the assay using a Shimadzu 1240 UV-visible spectrophotometer (Kyoto, Japan). The peptides were dissolved in PBS, and 10 µL aliquots of each peptide solution were incubated with 190 µL of the ABTS^+^ solution in triplicate, at final concentrations of 0.05, 0.025, and 0.0125 mg/mL. After a 1 h incubation in the dark, the absorbance was measured at 734 nm. The radical-scavenging activity was quantified against a standard curve, with the results expressed as milligrams of Trolox equivalents per milligram of peptide.

### 2.5. Interaction of Peptides with Lipid Membranes

#### 2.5.1. Monolayers at the Air/Water Interface

Peptide adsorption and insertion into lipid monolayers was determined using a Langmuir NIMA 102M balance (NIMA Technology, Coventry, UK).

To assess peptide adsorption at the air/water interface, a cuvette containing 1 mL of a 10 mM HEPES buffer solution with 150 mM NaCl, pH 7.4, was utilized. Aliquots of a concentrated peptide solution were injected in the buffer through a hole in the cuvette wall, reaching final peptide concentrations of 8, 32, and 64 μM. The time evolution of the surface pressure (π) was registered after each injection until a plateau was reached.

The incorporation of the peptides into lipid monolayers was evaluated using the same setup, preparing the lipid monolayer before peptide injection. For this, small drops of a chloroformic solution of the total lipid extract of *E. coli* (Avanti Polar Lipids, Inc.; Alabama, USA) were spread on the buffer surface until a surface pressure of 30 mN/m was reached. After evaporating the chloroform for 5 min, peptide aliquots were injected into the buffer through the cuvette hole and π was measured over time. The change in π was quantified as
(5)$$\Delta \pi =\pi_{f}-\pi_{o}$$where π_f_ is the final surface pressure (value at plateau) and π_0_ is the initial surface pressure. All experiments were conducted at a stable room temperature of 25 ± 3 °C.

#### 2.5.2. Large Unilamellar Vesicles (LUVs)

Large unilamellar vesicles (LUVs) were prepared with the total lipid extract of *E. coli* (Avanti Polar Lipids). For this, an aliquot of 1 mg of lipids dissolved in chloroform was deposited into a glass tube and dried by evaporation with nitrogen (N_2_). The glass tube underwent vacuum treatment to eliminate any residual solvent traces. The lipids were then hydrated in 200 μL of a 5,6-carboxyfluorescein (CF) solution (Sigma-Aldrich) at a concentration of 49 mM, pH 8, and 431 mOsm/kgH2O through nine cycles of heating (70 °C for 30 s), vortexing (1 min), and freezing (ice bath for 30 s). A polycarbonate filter with a pore size of 100 nm was employed to standardize the liposome size. Twenty-one extrusion cycles were conducted at 40 °C through the filter, resulting in LUVs with an average diameter of 100 nm. Size exclusion chromatography (Sephadex G-25) was utilized to remove the CF outside of the LUVs.

#### 2.5.3. Fluorescence Excitation Spectra of 5,6-Carboxyfluorescein (CF)

The fluorescence spectra of the CF-filled LUVs were measured using a FluoroMax3 Jobin Yvon spectrofluorometer (Horiba Jobin Yvon; Edison, NJ, USA). A 200 μL aliquot of the LUVs containing 100 μM CF was utilized for the experiment. Excitation was performed over a wavelength range of 400 to 540 nm, with the emission recorded at 517 nm. The peptides were added at various final concentrations (8, 32, 50, and 64 μM), and the fluorescence was assessed 10 min after each addition. The percentage of CF release was calculated using the fluorescence intensity at 517 nm with an excitation wavelength of 490 nm, using the following equation:
(6)$$\%CF=100\times \left[\frac{I_{t}-I_{0}}{I_{\infty }-I_{0}}\right]$$ where I_t_, I_0_, and I_∞_ represent the fluorescence intensities at a specific time point, at the beginning of the experiment, and after the addition of 10% *v*/*v* Triton X-100, respectively. Duplicate experiments were performed for each condition.

#### 2.5.4. Dynamic Light Scattering (DLS)

The size distribution of the LUVs was assessed using a nanoparticle analyzer (Horiba, SZ-100) through dynamic light scattering (DLS) analysis. The measurements were conducted at 300 kHz for 7 min at 25 °C. For each run cycle, 200 μL of the LUV solution (100 μM) was placed in a glass tube, and aliquots of the peptide stock solution were added to achieve final concentrations of 8, 32, 50, and 64 μM. The size of the vesicles was determined before and after the addition of each peptide concentration.

## 3. Results

### 3.1. Selection of Peptides and Sequence Analysis

The interactions between AMPs and cell membranes are dictated by various molecular properties, including hydrophobicity, net charge, secondary structure, and amphipathicity [39]. To enhance the bioactivity of the parent peptide hylin-Pul3, which was originally isolated from the skin of the frog *Boana pulchella* [20], a series of derivative peptides were rationally designed. The net positive charge was increased by substituting specific amino acids with cationic residues, such as Arg (R) and Lys (K). The hydrophobicity was augmented by introducing residues like Trp (W), while modifications were implemented to optimize amphipathicity and the spatial distribution of cationic amino acids along the helical structure (Figure 1, Step 2).

A total of 1064 peptide sequences were designed based on the primary structure of hylin-Pul3. These modifications targeted specific regions of the naturally occurring peptide. Using in silico bioactivity prediction, a rigorous screening process was conducted to identify promising candidates, focusing on peptides with predicted probabilities exceeding 0.85 according to the consensus results from CAMP_R3_ software. This screening was further complemented with evaluations using DBAASP and hemolytic activity predictions through DBAASP v3, HAPPENN, and HemoPred software (Figure 1, Step 3).

Six derivative peptides were selected for synthesis and subsequent in vitro evaluation (Figure 2 and Appendix A). Of these, five peptides retained the original length of 19 amino acids, while peptide dHP3-31 was truncated to 15 amino acids. The net charge of the derivatives ranged from +6 to +8, representing a substantial increase compared to the parent peptide. All derivatives were predicted to adopt an α-helical conformation, with a helicity exceeding 60%. The predicted three-dimensional α-helical structures are depicted in Figure 3.

The enhancement of net positive charge in the designed peptides resulted in a concurrent decrease in hydrophobicity relative to the parent peptide. However, the hydrophobic ratio remained within the range of 47 to 60%, consistent with the characteristics of the native peptide (see Appendix A).

Heliquest v1.2 online online software was employed to perform a helical wheel projection analysis of the amphipathic peptides (Figure 2). The selected derivative peptides exhibited a clearly defined amphipathic structure. Specifically, peptides dHP3-50, dHP3-50.190, dHP3-50.137, and dHP3-31 were designed to exhibit a broad dispersion of cationic residues on the polar face of the helix, whereas peptides dHP3-84 and dHP3-84.39 displayed a more concentrated distribution of charged amino acids within the polar region.

Additionally, charged polar amino acids were introduced into peptides dHP3-50.190 and dHP3-84.39 to disrupt the nonpolar region of the helix, resulting in “imperfect amphipathic” peptides. Similarly, peptide dHP3-50.137 was engineered with a disruption in the nonpolar region, achieved through the incorporation of a neutral residue (Ala). Peptides dHP3-31 and dHP3-50.137 also exhibited increased hydrophobicity due to the incorporation of a Trp residue within the nonpolar region of the amphipathic helix.

### 3.2. Synthesis of Peptides

The selected peptides were all manually synthesized, purified by chromatographic procedures, and their sequences were determined and confirmed by mass spectrometry, as detailed in the Appendix A. The sequences of hylin-Pul3 and its derivative peptides are presented in Figure 2.

### 3.3. Improved Biological Activity

Through rational design, we successfully enhanced the antimicrobial activity against Gram-negative strains by deriving peptides from an active AMP. Peptide dHP3-50.137 demonstrated a reduction in MIC of up to 8-fold, while peptides dHP3-31, dHP3-50, dHP3-50.137, and dHP3-84 exhibited a 4-fold reduction against *E. coli* and *A. baumannii* strains compared to the parent peptide hylin-Pul3. Peptide dHP3-84 also achieved a 4-fold lower MIC than the original peptide against the *K. pneumoniae* strain.

In contrast, for the Gram-positive strains *S. aureus* and *S. mutans*, the MIC values of all designed peptides—except dHP3-84—increased by 2-fold to 8-fold relative to that of the natural peptide hylin-Pul3.

Synergy studies between the derived peptides dHP3-84 and dHP3-50.137 against the *E. coli* strain revealed a synergistic interaction, as indicated by a fractional inhibitory concentration index of 0.5. The combined activity showed a 2-fold improvement compared to the activity of each peptide individually. The antimicrobial and synergistic activity results are presented in Table 1.

To evaluate the antiviral activity of the peptides hylin-Pul3 and dHP3-84, they were tested against *Varicellovirus suidalpha1* virus. The experiments were conducted at non-cytotoxic concentrations that were previously established for Vero cell monolayers. The antiviral assays performed at different stages of virus replication revealed a 70% inhibition for dHP3-84, whereas hylin-Pul3 did not exhibit any inhibitory activity (Appendix A).

### 3.4. Hemolytic Activity

The hemolytic activity of the peptides was assessed by quantifying their ability to lyse human red blood cells at varying concentrations. The hemolytic potential was categorized based on established criteria: samples inducing <5% hemolysis were deemed non-hemolytic [40]. Peptides dHP3-31, dHP3-50, and dHP3-50.190 did not exhibit detectable hemolytic activity at the highest concentration tested (200 µg/mL; equivalent to 115.33 µM, 97.16 µM, and 92.7 µM, respectively). For peptides displaying measurable hemolytic activity (hylin-Pul3, dHP3-50.137, dHP3-84, and dHP3-84.39), the concentration required to induce 2% hemolysis (HC2) was determined. For all the derivative peptides, the HC2 values exceeded 39 μM, supporting their classification as non-hemolytic molecules (see Table 1).

### 3.5. Cytotoxic Activity

The cytotoxic effects of hylin-Pul3 and its derivatives (dHP3-31, dHP3-50, dHP3-50.137, dHP3-50.190, dHP3-84, and dHP3-84.39) were evaluated on human fibroblast cells. As depicted in Figure 4, significant reductions in fibroblast viability were observed with 1 μg/mL of both hylin-Pul3 and its derivative dHP3-31 (0.5 and 0.50 µM, respectively). For peptide dHP3-50, cytotoxic effects were detectable at a concentration of 10 μg/mL (4.9 µM), whereas dHP3-50.137 exhibited toxicity starting at 25 μg/mL (12.4 µM). Peptide dHP3-84 only displayed cytotoxic effects at the highest concentration tested (100 μg/mL; 46 µM).

Interestingly, peptide dHP3-84.39 demonstrated a unique response: the fibroblast viability increased at 5 μg/mL (2.3 µM), potentially suggesting a proliferative effect. However, at higher concentrations (50 μg/mL; and 100 μg/mL), a decline in viability was observed. No significant cytotoxic effects were detected for the dHP3-50.190 peptide across all the concentrations tested.

The cytotoxic effects of hylin-Pul3 and dHP3-84 on Vero cells were also evaluated to determine the peptide concentrations that do not disrupt the cell monolayer, thereby facilitating an accurate assessment of antiviral activity. The Maximum Non-Cytotoxic Concentration (MNCC) was established as 25 μg/mL (13.5 µM) for hylin-Pul3 and 125 μg/mL (57.6 µM) for dHP3-84 (Figure 5, Appendix A). Cultures exposed to concentrations below these thresholds maintained a morphology comparable to the untreated controls, whereas higher concentrations resulted in visible disruption of the cell monolayer.

### 3.6. In Vitro Antioxidant Assays

The ABTS assay demonstrated moderate antioxidant activity for the derivative peptides dHP3-50.137 and dHP3-31, with values of 0.08 and 0.14 mg Trolox equivalents per mg peptide, respectively. In contrast, the DPPH assay did not reveal significant activity for these peptides (Appendix A).

### 3.7. Interaction with Lipid Membranes

#### 3.7.1. Monolayers at the Air/Water Interface

The surface activity of the peptides hylin-Pul3, dHP3-50, dHP3-50.137, dHP3-84, and dHP3-84.39 was evaluated by introducing the peptides at final concentrations 8, 32, and 64 μM into a buffer and monitoring the changes in π. The accumulation of the derived peptides at the air/water interface caused surface pressure changes (Δπ) ranging from 8 to 15 mN/m (Figure 6a). In contrast, hylin-Pul3 exhibited a Δπ of 33 mN/m at 8 μM, which increased to approximately 38 mN/m at 64 μM. This is in agreement with the higher hydrophobicity and hydrophobic ratio (leading to a higher influence of the hydrophobic effect) and lower charge (leading to a lower intermolecular repulsion at the interface) of this peptide compared to the derivatives (see Appendix A).

To further assess the peptides’ ability to insert into a lipid film, Δπ was evaluated in the presence of a lipid monolayer using the total lipid extract of *E. coli* at 30 mN/m, which has a lipid density comparable to that of a bilayer. The results are presented in Figure 6b, where it can be observed that the natural peptide hylin-Pul3 promoted the highest changes in the film surface pressure.

This is consistent with the high tendency shown by hylin-Pul3 to accumulate at the bare interface, which indicates that this peptide is a strong surfactant. However, the final surface pressure reached by the peptide in the absence or presence of lipids was similar (38 mN/m in a clean interface and 40–42 mN/m in the presence of lipids). In other words, the peptide did not specifically interact with the lipids. In contrast, the derivatives reached surface pressures of 34–38 mN/m in the presence of lipids, which were 20–30 mN/m higher than those reached at the bare interface. This indicates that the derivatives have affinity for the lipid monolayer beyond that related to the surface activity of the peptides.

Notably, dHP3-50.137, being one of the less surface-active peptide, induced the highest monolayer perturbation compared to the other derivatives. In the other extreme, dHP3-84 produced the highest Δπ at the bare interface but only increased the surface pressure of the lipid monolayers by about 5 mN/m, thus inducing a weak film perturbation effect. The described peptide behaviors cannot be directly related with the peptide hydrophobicity or hydrophobic moment (Appendix A). The presence of the Trp residue in dHP3-50.137 may be responsible for the strong film perturbation effect induced by this peptide.

#### 3.7.2. Peptide Activity on Large Unilamellar Vesicles (LUVs)

To further investigate the interaction between the peptides and lipid bilayers, liposomes containing the carboxyfluorescein (CF) fluorophore were employed. The fluorescence spectra of CF were measured in the presence and absence of the peptides hylin-Pul3, dHP3-84, dHP3-50, dHP3-50.137, and dHP3-89.39 to assess if they can induce membrane permeabilization or lysis. Under control conditions, CF fluorescence is self-quenched within the vesicular compartment; thus, an increase in fluorescence indicates leakage of CF molecules from the vesicle lumen [41]. Additionally, the size distribution of vesicles composed of the *E. coli* lipid extract was determined before and after the addition of the peptides (see Appendix A).

Figure 7 shows the results for the release of CF from LUVs. Strikingly, dHP3-84 did not promote the release of CF at all the tested peptide concentrations, suggesting that this peptide’s action is not related with an increase in membrane permeability. The peptides dHP3-50.137, and dHP3-84.39, on the other hand, exhibited similar behaviors, losing 50% of the CF at the lowest peptide concentrations. Hylin-Pul3 and dHP3-50 depicted intermediate behaviors.

The sensitivity of the LUVs’ permeability to dHP-50.137 is in agreement with the perturbation induced by this peptide on lipid monolayers and with the low MIC against *E. coli*.

The addition of the peptides resulted in the formation of aggregates with a diameter of 400 nm or larger for all peptides except for hylin-Pul3 and dHP3-84 (see Appendix A). This increase in the average size can be attributed to vesicle aggregation due to neutralization, which is expected when cationic peptides adsorb to anionic vesicles (total lipid extracts from *E. coli* contain 15% phosphatidylglycerol and 10% cardiolipin). Following this hypothesis, the lack of aggregation in the experiments with hylin-Pul3 may be due to the lower peptide charge. The same result with dHP3-84 may be due to the low affinity of the peptide to the membrane, as was suggested by the CF leakage results and the small increase in the film surface pressure caused by this peptide.

## 4. Discussion

Numerous studies on structure–activity relationships (SAR) emphasize the importance of the physicochemical properties of peptides for their antimicrobial activities. Modifications to their properties such as size, hydrophobicity, charge, helical content, and amphiphilicity can significantly influence the potency and spectrum of α-helical AMPs [10,11].

Most AMPs are cationic, with a higher net positive charge often associated with increased antimicrobial activity [42,43,44]. The natural peptide hylin-Pul3 (net charge of +3) was compared with selected derivative peptides with increased charges (+6 to +8). Our results suggest that this increase in cationicity correlates with enhanced antimicrobial activity, particularly against Gram-negative strains such as *E. coli* and *A. baumannii*. This trend could be attributed to the distinct structural characteristic of their cell membranes. Gram-negative bacteria, due to their complex structure, require stronger electrostatic interactions for effective AMP activity, while Gram-positive bacteria, with more permeable membranes, can interact efficiently with peptides with lower cationic charges [45,46]. Among peptides with the same net charge, those with a higher proportion of arginine showed greater antimicrobial activity and broader-spectrum efficacy. This effect is likely due to the guanidinium group of arginine, which provides a higher positive charge density, facilitating more effective membrane permeabilization than lysine-containing peptides [47].

Linear amphipathic α-helices are among the most common structures in AMPs [10]. This structural arrangement allows the hydrophilic side to interact with the head groups of anionic phospholipids, while the hydrophobic side integrates into the lipid bilayer core [48]. A high amphipathicity, indicated by a high hydrophobic moment, correlates with increased antibacterial activity, primarily against Gram-negative bacteria [49]. In our study, the high amphipathicity, indicated as a hydrophobic moment, of derivative dHP3-84 (Appendix A) correlated with enhanced antibacterial efficacy, particularly against Gram-negative bacteria. Although some studies have suggested that disrupted amphipathicity can improve efficacy [15,50], our results indicate otherwise. Peptides dHP3-50.190 and dHP3-84.39 exhibited weaker antimicrobial activity compared to their counterparts dHP3-50 and dHP3-84, which had perfect amphipathicity.

To assess the role of aromatic residues in antimicrobial activity, dHP3-50 was modified by incorporating a Trp residue on its nonpolar face to create dHP3-50.137, and by removing four N-terminal residues to create dHP3-31. These modifications improved the activity against *E. coli* and *A. baumannii* while maintaining efficacy against *K. pneumoniae* (Table 1) but resulted in reduced effectiveness against Gram-positive strains. The enhanced activity of dHP3-50.137 may be due to its higher membrane binding affinity, influenced by the unique properties of the Trp residue, as documented in other studies [51,52] and revealed here from the monolayer penetration and LUV leakage experiments. The reduced efficacy of dHP3-31 may be attributed to its shorter length, which limits its ability to fully penetrate the lipid bilayer [53].

Combining AMPs can enhance bacterial inhibition while reducing the risk of resistance development through cooperative mechanisms [54]. This strategy also reduces the required peptide dosage, thereby lowering the associated costs [55]. The synergy observed between dHP3-84 and dHP3-50.137 against *E. coli* suggests complementary membrane interactions. Specifically, dHP3-50.137 may compromise membrane integrity, facilitating the uptake and intracellular access of dHP3-84. However, this mechanism was not fully explored in the present study.

Peptides derived from amphibian skin have shown potent antiviral activity against *Varicellovirus suidalpha1* [56,57,58,59]. Here, dHP3-84 exhibited a significant antiviral effect, inhibiting 70% of HSV-1 infectious particles, whereas hylin-Pul3 did not show any measurable activity. The optimized cationic and amphipathic properties of dHP3-84 likely enhanced its efficacy by causing viral envelope disruption, inhibiting host cell adhesion, and/or suppressing viral replication [60].

Many natural AMPs have moderate antimicrobial activity but high cytotoxicity, making peptide modification a valuable approach to enhance their therapeutic potential [61]. Our derivative peptides showed low hemolytic activity at the tested concentrations, indicating higher selectivity for bacterial cells compared to hylin-Pul3. Reducing the hydrophobicity resulted in decreased hemolysis relative to the natural peptide, aligning with the results of previous studies [62,63,64]. In the case of peptide dHP3-84.39, the incorporation of polar amino acids on the nonpolar face disrupted the ideal amphipathicity, leading to a marked reduction in hemolytic activity.

Fibroblast cells serve as a sensitive model for assessing cytotoxicity [65]. Here, significant cytotoxic effects were observed for hylin-Pul3, dHP3-31, dHP3-50, and dHP3-50.137 at concentrations below 25 μg/mL, whereas dHP3-84 exhibited cytotoxicity only at higher concentrations. Interestingly, dHP3-84.39 showed an unusual response, potentially indicating a proliferative effect at low concentrations. The differences in cytotoxicity likely resulted from variations in the cationic charge density and the distribution of polar residues on the peptide surface [65].

AMPs are widely recognized for their potential in mitigating oxidative stress-related damage [66,67]. In our study, dHP3-50.137 and dHP3-31 showed notable antioxidant activities in the ABTS assay, likely due to the presence of tryptophan. The ability of tryptophan to stabilize free radicals through its resonance structure might enhance the radical-scavenging capacity of tryptophan-containing peptides [68,69]. The differences between the radical-scavenging activities measured with the DPPH and the ABTS assays suggests that these peptides may be more effective in aqueous environments, highlighting their potential as antioxidants in biological systems. The superior performance of the shorter peptide dHP3-31 suggests that peptide length, which influences residue exposure and structural flexibility [70], may be an important factor in optimizing antioxidant efficacy.

Additionally, the peptide–lipid interactions revealed distinct mechanisms of action. Hylin-Pul3 induced the leakage of vesicle contents without changes in the vesicle size, whereas dHP3-84.39, dHP3-50, and dHP3-50.137 promoted vesicle aggregation. dHP3-84 did not affect bilayer permeability and induced the lowest changes in the surface pressure of the lipid films. Therefore, our findings indicate that dHP3-84 may operate through intracellular mechanisms, rather than through direct membrane disruption.

Machine learning (ML) models have demonstrated their value in predicting peptide bioactivity [17,71]. The derived peptides dHP3-84, dHP3-50.190, and dHP3-84.39 outperformed hylin-Pul3 in terms of enhanced antimicrobial activities and reduced cytotoxicity on eukaryotic cells, which, taken together, indicates higher selectivity and therapeutic indices. The integration of bioinformatics tools with rational design offers significant potential for improving peptide efficacy while minimizing adverse effects [19]. Despite these advancements, challenges remain in expanding datasets to enhance the predictive accuracy of computational models, as well as in overcoming the limitations of peptides to successfully translate promising in vitro results to actual in vivo applications [72].

While not all designed peptides are immediate candidates for use as antibiotics in the pharmaceutical industry, they could play a key role as adjuvants to conventional antibiotics or they may find applications in the food industry. The results obtained require further analysis to enhance the peptides’ stability and bioavailability, with a particular focus on dHP3-84, aiming to enhance its performance against Gram-negative strains. In this regard, the rational design of shorter, cyclic, or stapled peptides could be an interesting avenue to explore in order to improve their pharmacokinetic properties while maintaining or even enhancing their antimicrobial efficacy.

## 5. Conclusions

Advances in bioactive peptide design hold great potential for the development of next-generation therapeutic agents, particularly to combat antimicrobial resistance. In this study, our rationally designed peptides outperformed the natural AMP template in both bioactivity and selectivity. Among the six derivatives tested, dHP3-84 displayed broad-spectrum antimicrobial activity, including synergistic interactions with dHP3-50.137 against *E. coli*, as well as significant antiviral activity against *Varicellovirus suidalpha1*. All the analogs showed reduced cytotoxicity compared to hylin-Pul3. Additionally, peptides dHP3-31 and dHP3-50.137 exhibited moderate antioxidant activities, further expanding their therapeutic potential. The analysis of the derivatives’ interactions with lipid bilayers revealed varying degrees of membrane permeabilization, suggesting distinct mechanisms of action compared to the parent peptide.

These findings highlight the effectiveness of rational design in optimizing peptide functionality. SAR studies were instrumental in guiding the design process, while computational tools enabled the efficient screening and classification of the AMPs. This approach facilitated the identification of sequences with superior bioactivity and significant therapeutic potential.

## Figures and Tables

**Figure 1 biomolecules-15-00449-f001:**
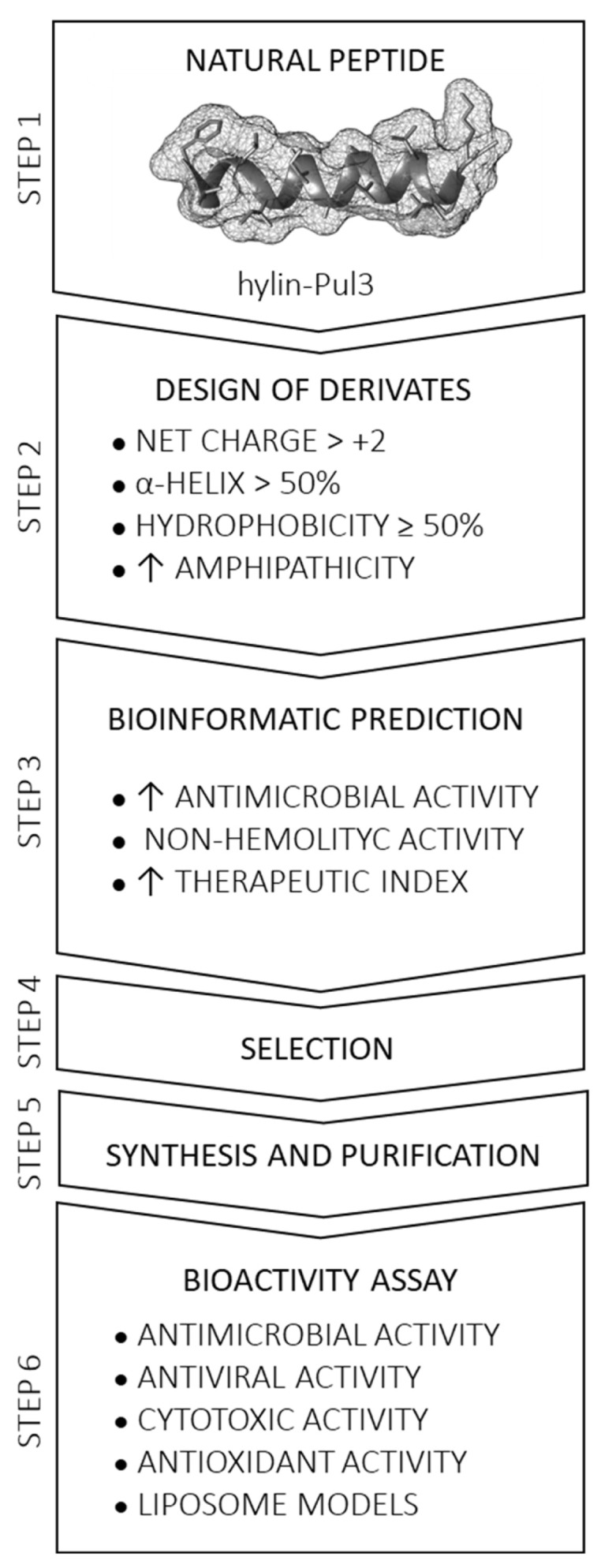
Workflow for the design and selection of optimized peptide derivatives from a natural peptide template. Step 1: Selection of a peptide sequence with established antimicrobial activity. Step 2: Rational modification of physicochemical properties. Step 3: Prediction of biological activity using in silico tools. Steps 4–6: Selection of peptides with desired characteristics, synthesis, purification, and in vitro testing to evaluate their bioactivity.

**Figure 2 biomolecules-15-00449-f002:**
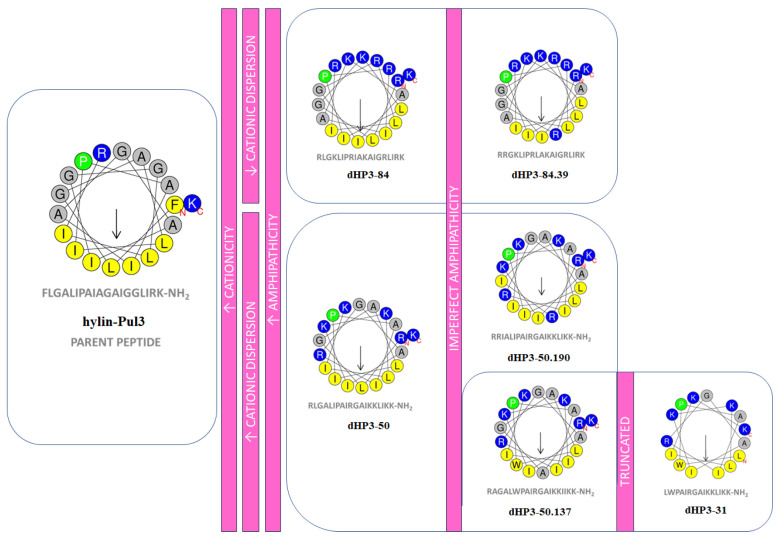
Structural modifications of hylin-Pul3 derivatives. Helical wheel projections of hylin-Pul3 and its rationally designed derivatives, highlighting structural changes aimed at enhancing biological activity. Residues are color-coded: nonpolar hydrophobic residues are yellow, polar basic residues are blue, glycine and alanine are gray, and proline is green. Arrows represent the direction and magnitude of the hydrophobic moment (µH) and the residue marked with “N” is the N-terminal end of the amphipathic helix and the residue marked “C” is the C-terminal.

**Figure 3 biomolecules-15-00449-f003:**
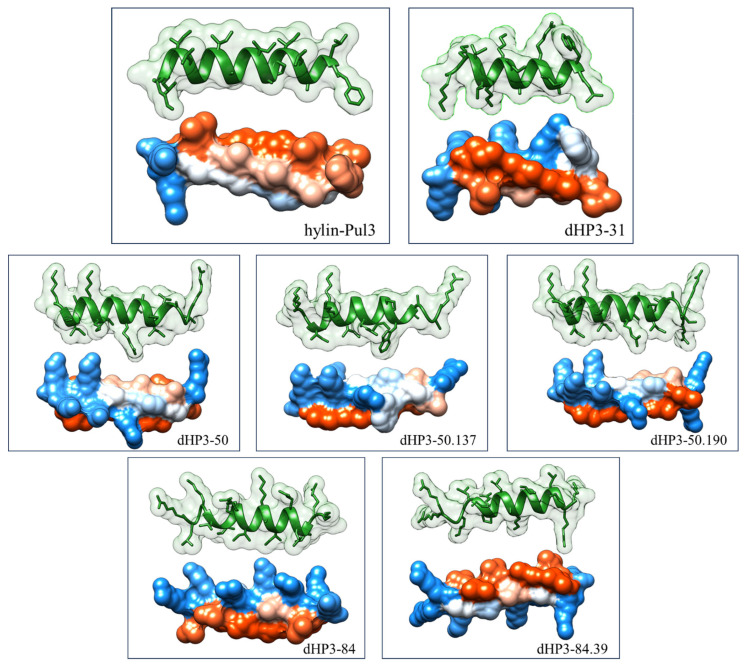
Three-dimensional α-helical structures of hylin-Pul3 and its derivatives peptides. Top: ribbon diagram of the peptide, with its accessible surface area represented. Bottom: 3D structure of the peptide with a hydrophobicity surface map (red: hydrophobic residues; blue: hydrophobic residues). The structures were predicted using AlphaFold2 and visualized with UCSF Chimera.

**Figure 4 biomolecules-15-00449-f004:**
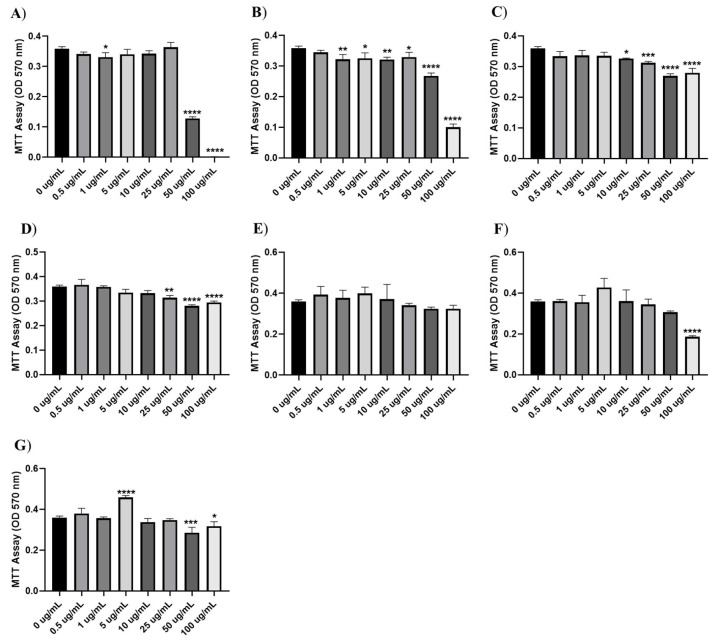
Cytotoxic effects of antimicrobial peptides (**A**) hylin-Pul3, (**B**) dHP3-31, (**C**) dHP3-50, (**D**) dHP3-50.137, (**E**) dHP3-50.190, (**F**) dHP3-84, and (**G**) dHP3-84.39 in human fibroblast cells at concentrations of 0.5, 1, 5, 10, 25, 50, and 100 μg/mL. (One-Way ANOVA, Dunnet’s multiple comparisons test were used for statistical analysis. *p* < 0.05 were considered statistically significant. In this way, * *p* < 0.05, ** *p* < 0.01, *** *p* < 0.001 and **** *p* < 0.0001).

**Figure 5 biomolecules-15-00449-f005:**
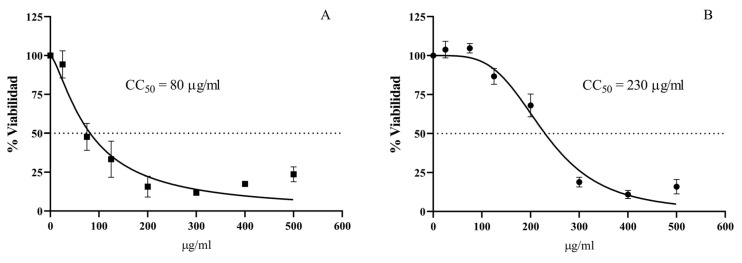
Cytotoxic effect of (**A**) hylin-Pul3 and (**B**) dHP3-84 peptides on Vero cell monolayers. The figure displays the concentrations (µg/mL) at which each peptide induces 50% cell death. The corresponding values in µM are provided in Appendix A.

**Figure 6 biomolecules-15-00449-f006:**
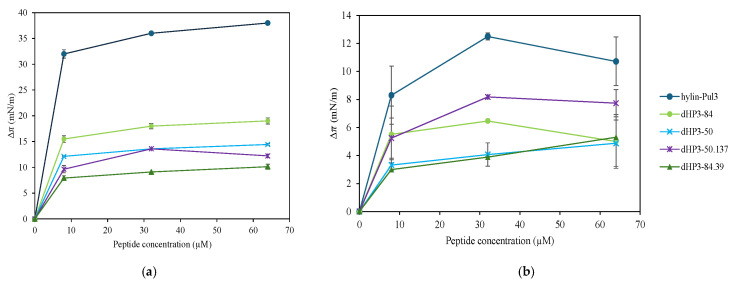
Surface pressure variation at different peptide concentrations (8, 32, and 64 μM). (**a**) In the absence of lipid films. (**b**) In the presence of lipid films, with an initial surface pressure of 30 mN/m. The increase in surface pressure indicates the peptides’ ability to integrate into the monolayer.

**Figure 7 biomolecules-15-00449-f007:**
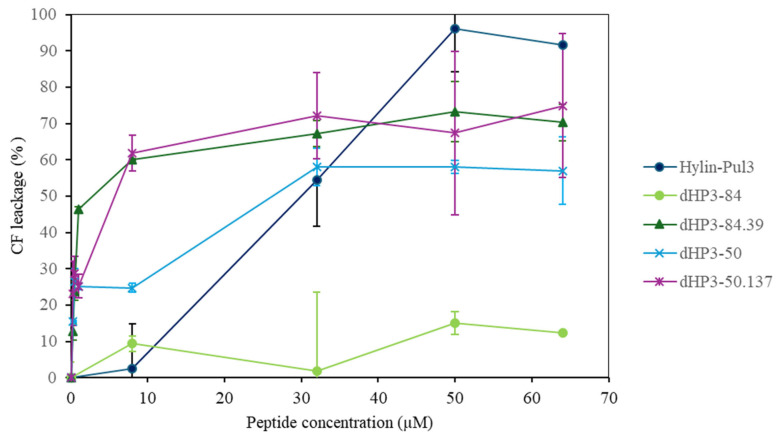
Percentage release of the soluble fluorophore carboxyfluorescein (CF) as a function of peptide concentration. CF leakage is indicated by an increase in fluorescence at ex 490 nm/em 517 nm.

**Table 1 biomolecules-15-00449-t001:** Determination of antimicrobial, synergistic, and hemolytic activities of hylin-Pul3 and its derivative peptides. MIC: minimal inhibitory concentration.

Name	Antimicrobial Activity (MIC, µM)					Synergistic Activity	Hemolytic Activity (2%) (µM)
	*E. coli*	*S. aureus*	*K. pneumoniae*	*A. baumannii*	*S. mutans*	*E. coli* (ATCC 25922)	
	*ATCC 25922*	*ATCC 25923*	*ATCC 13883*	*ATCC 13304*	*ATCC 25175*	FIC Index	
hylin-Pul3	32.0 ± 0.0	7.3 ± 1.6	>64	32.0 ± 0.0	4.6 ± 1.6		10
dHP3-31	16.0 ± 0.0	37.3 ± 13.1	>64	9.3 ± 3.2	18.7 ± 6.5		>115
dHP3-50	8.0 ± 0.0	18.6 ± 6.5	64.0 ± 0.0	6.6 ± 2	8.0 ± 0.0		>97
dHP3-50.137	4.0 ± 0.0	64.0 ± 0.0	53.3 ± 16.5	8.0 ± 0.0	32.0 ± 0.0	0.5 (Synergy)	221
dHP3-50.190	9.3 ± 3.2	29.3 ± 6.5	>64	16.0 ± 0.0	32.0 ± 0.0		>93
dHP3-84	8.0 ± 0.0	9.3 ± 3.3	18.6 ± 6.5	8.0 ± 0.0	7.3 ± 1.6	0.5 (Synergy)	39
dHP3-84.39	6.6 ± 4.8	26.6 ± 8.3	64.0 ± 0.0	16.0 ± 0.0	>64		376

## Data Availability

Data are contained within the article and Appendix A.

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
