# Peer review of "Enhancing Antimicrobial Peptides from Frog Skin: A Rational Approach"

_biomolecules, 2025, doi:10.3390/biom15030449_

Round 1

Reviewer 1 Report

Comments and Suggestions for Authors

This is a really consistent and sound study and the presented results excatly reflect the title of this manuscript: how to come from a model AMP sequence to optimized sequences with special respect on potency and selectivity. This rational approach is really clear presented and was really successful. 

I have only minor comments:

1. Why HSV-1 was chosen as an "antiviral" model ? What is the rationale behind this. Is this a representavive virus ? Please comment on this.

2. The authors should comment, why 15-19 amino acid length is neccessary for a potent AMP ? Shorter variants would be desired and much easier to produce, cheaper and most probably with higher plasma stability. Further derivatization e.g. cyclization may lead to smaller peptides with same or better pharmacological properties. Please comment this point, maybe as a perspective.

3. Using your process you came from > 1000 sequences to 6 candidates. What time point in the selection process you consider as the weakest that that could be optimized ?

4. Missing somehow the perspective: in which direction will the authors go with the rational approach or with one of the analyzed peptides ?

5. Please comment on the purity of the peptides after synthesis (supplemental) in %. It should be 95% or more. E.g. peptide in Figure S3 may not fullfill this criteria.

6. Especially the numbering in the MS spectra (S1 to S7; supplemental) is too small, it is impossible to read. Most of these numbers could be deleted I think.

Reviewer 2 Report

Comments and Suggestions for Authors

The authors present the results of SAR-type studies on the structure and biological activity of the antimicrobial peptide Hylin-Pul3 and its analogs. The topics are interesting and potentially utilitarian-associated with the possibility of using peptides as compounds with bacterial activity. Unfortunately, the manuscript contains a fair number of defects that need to be corrected/clarified before possible publication. Detailed list of perceived errors below.

30 In the Introduction section, there is no information about the origin of the parent peptide

116-139 description of the tools used for the study is duplicated (sections116-127 and 128-139)

143 Fig 1. The factual/scientific value of this figure is low. Suggestion, either remove it completely or leave it in text form only, without pictograms and colorful embellishments. The SPPS pictogram is completely illegible.

159 minimal power -please provide value  ?W

163  data missing: ?W, coupling time?, temp? single coupling?

165 deprotection time? The process was conducted once or twice?

167 (95:….) v/v/v

168 It is not recommended to lyophilize peptides from acetonitrile solutions (due to the fact that ACN freezes at low temp., has a bad effect on the vacuum in the lyophilizer and the pump), especially such concentrated ones.

169 5 mg of synthetic… purity?

171 Reverse not reversed

403 Fig. 2 Some symbols are not defined/explained: an arrow, pink N,C-letters

417 Fig3 These drawings (both the top and bottom figure for a particular peptide) show much more than just the 3D structures of the peptides. Please include a more detailed description for this figure.

444 A.Baumannii -italic

457 Table 1 No standard deviation values

458, 503 There is no antiviral and antioxidant data presented.

463  Why is the title of subsection 3.4 Cytotoxic Activity, and it describes the hemolytic properties of peptides? Subsection 3.5 again describes cytotoxic properties.

Notes to Support data section

From the description of the HPLC chromatograms, it is impossible to know whether these are crude peptides after removal from the resin or after purification. (A synthetic peptide can come from either step). If it is the latter, they are unsuitable for any study in this form due to low purity. Please attach the HPLC of the purified peptides used for the study.

From the description of the HPLC chromatograms (Fig S_(A)), it is impossible to know whether these are crude peptides after removal from the resin or after purification. (A synthetic peptide can come from either step). If it is the latter, they are unsuitable for any study in this form due to low purity. Please attach the HPLC of the purified peptides used for the study.

(Fig S_(B)) unless MALDI TOF/TOF not TOF

Comments on the Quality of English Language

I don't feel qualified to judge the quality of the English language.

Reviewer 3 Report

Comments and Suggestions for Authors

The work is interesting and provides interesting results from the big amount of performed experiments. I have some remarks more developed in the attached file.

Round 2

Reviewer 2 Report

Comments and Suggestions for Authors

I accept the authors' explanations and the corrections made. I think that the revised version can be recommended for printing I have only 2/3 small comments.
p 30. I noticed that in the Introduction section the authors state the origin of the peptide. However, I think that such information should already be in the Abstract section. The Abstract is often a pseudo-self-contained section representing the entire publication (e.g., in the PubMed database). If this information is not there, the reader will not know what he is dealing with. The name will be there, but the origin will not. And it is known that interspecies peptide/protein sequences are not 100% conservative. That is, so it will not be entirely clear what was studied. However, I would recommend adding this information to the Abstract section. 

p 457 Why are only some MICs values given a standard deviation? Providing values without a deviation looks suspicious to say the least.

Regarding the data in the Support section.
Indeed, showing a good MS spectrum containing peak(s) derived solely from the peptide effectively means that it is “clean.” However, its purity cannot be stated on this basis. Possible impurities undergoing much weaker ionization may not be visible. A good working standard for peptide studies is to have an HPLC chromatogram of the peptide after purification and lyophilization. 
